# Semi-supervised Abdominal Multi-Organ and Tumors Segmentation by Cascaded nnUNet

Bochen Wu[1][*][0009−0000−7216−2541], Mengyao Zhang[1][*], and Wenli Fu[1][†][0000−0001−7059−3495]

Shanghai Jiao Tong University, Shanghai, China
{LilyFu}@sjtu.edu.cn

**Abstract.** Abdominal multi-organ and tumors segmentation can provide anatomical structure information for doctors and is an important step in computer-aided diagnosis. However, accurate segmentation of abdominal multi-organ and tumors is still an urgent problem due to partially labeled issue and variable tumor position. To address these problems, we propose a cascaded approach using cascaded nnU-Net to handle the task of multi-organ and tumors segmentation. Since tumors located in different organs have different gray value and textures, we train segmentation models for each tumor to improve the tumor segmentation accuracy. We also combine semi-supervised method while training to makes full use of the unlabeled data. In addition, we postprocess the segmentation results to refine segmentation based on anatomical prior knowledge. We improve the inference speed by replacing the interpolation function and cropping the probability map. We obtain an average DSC of 90.28% on abdominal multi-organ segmentation and 42.87% on pan-tumor segmentation, with an average inference time of 23.77s per case on validation set.

**Keywords:** Semi-supervised learning · Multi-Organ segmentation · Tumor segmentation.

## 1    Introduction

Abdominal multi-organ segmentation is a fundamental task in the field of medical image analysis, providing crucial anatomical information for physicians and serving as a vital step in facilitating clinical diagnosis and surgical planning. However, due to variations in organ sizes and the prevalence of partially labeled organ datasets with significant differences between them, making automatic abdominal multi-organ segmentation remains a formidable challenge. Given that

---

* Equal Contribution

† Corresponding author

abdominal organs are frequently affected by tumors, accurate tumor segmentation is also necessary, which is crucial for early cancer detection, disease progression monitoring, intraoperative assistance, and treatment effect evaluation. However, the difficulty of tumor segmentation lies in the diversity of shape, size, and location of cancer lesions in different cases, as well as the blurred boundaries with healthy tissues, which make tumor segmentation more challenging. To address these problems, we propose a cascaded nnU-Net to handle abdominal multi-organ and tumor segmentation.

In terms of abdominal multi-organ segmentation, early studies mostly employed atlas-based methods[11,24], wherein the general framework involved deforming selected atlas images with segmentation structures onto the target image. However, in comparison to other body regions (e.g. brain), the abdominal region exhibits significant inter-subject variation, which seriously affects final accuracy. Recently, The Fast and Low-resource Semi-supervised Abdominal Organ Segmentation Challenge 2022 (FLARE22)[17] demonstrated that nnU-Net[9] can achieve excellent results in supervised learning, and when combined with pseudo labeling framework, it can attain state-of-the-art performance in semi-supervised tasks. Therefore, we also adopted the method of nnU-Net with pseudo labeling framework in our work.

The segmentation of tumors can be broadly categorized into two approaches in existing research: one involves training a separate segmentation model for each organ tumor and subsequently segmenting the tumor within the region of interest (ROI) of that particular organ; the other approach entails training a general model to segment all tumors in the entire abdominal imaging scan at once[12,23,2]. The latter method offers advantages in terms of model complexity and computing time, while the former method requires longer training and inference time. However, currently, the first method has achieved superior results compared to the second method because it performs segmentation on a smaller scale with individual models for each tumor. Therefore, we adopt the first method in our work and also utilize a pseudo labeling framework for tumor segmentation using unlabeled data.

In this paper, we propose a two-stage model for segmenting abdominal organs and tumors, along with an improved inference strategy based on nnU-Net to accelerate inference speed and reduce computational resources.

The contributions of this article can be summarized as follows:

– We employ pseudo-labeling-based semi-supervised learning for abdominal multi-organ and tumor segmentation, effectively utilizing unlabeled data.
– We introduce a coarse-to-fine segmentation framework that enhances tumor segmentation results at a fined scale.
– We leverage prior anatomical knowledge, and post-process the segmentation results to effectively minimize erroneous segmentation area.
– We replacing the interpolation function of nnU-Net and implementing GPU acceleration calculation as well as multi-process computation, which significantly accelerate the inference speed of our model.

## 2    Method

### 2.1    Preprocessing

We first crop the non-zero region of the image, then resample the cropped image to the median resolution of all data. Finally we normalize image using Z-Score normalization strategy. Z-Score normalization formula is as follows:

$$Z = x - \mu/\delta \tag{1}$$

$\mu$ is the mean of the CT values of the image foreground and $\delta$ is the variance of the CT values of the image foreground.

### 2.2    Proposed Method

Our method composes of two two 3D nnU-Net: Organ Segmentation Networks and Tumor Segmentation Networks, as can be seen in Fig1.

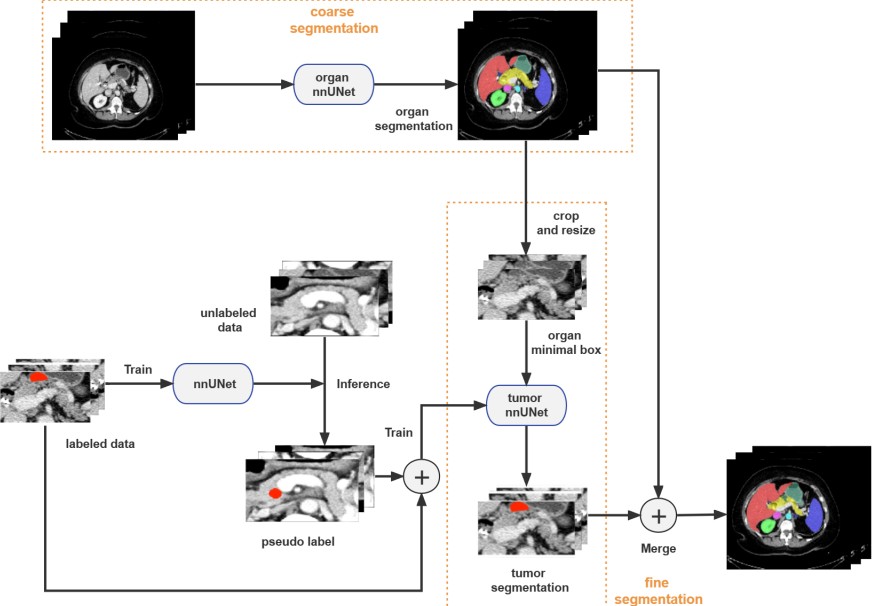

**Fig. 1.** Overview of cascaded nnU-Net framework. Organ segmentation model and tumor segmentation model are trained with pseudo-labeled and partially labeled data. First get the organ segmentation, then crop image according to organ mask to get the organ minimum box for tumor segmentation.

4        Wu et al.

We propose a coarse-to-fine frame which is commonly used in small-target segmentation task. The overall architecture of the method is shown in Fig1. It consists of organ segmentation network and tumor segmentation network. In the organ segmentation stage, we use nnU-Net network with the same setting as the FLARE22 best algorithm [8] to generate 13 abdominal organ segmentation masks. In the tumor segmentation stage, we crop the image according to organ mask obtained by organ segmentation net to get organ minimum box which is used as input in tumor segmentation network. In inference stage, we use the same strategy, i.e., segment organs first and then get tumor segmentation mask based on organ mask. Finally we merge organ masks and corresponding tumor masks to get final prediction.

**Loss function.** we combine the Dice Similariy Coefficient(DSC) loss and cross-entropy loss because compound loss functions have been proven to be robust in various medical image segmentation tasks [13].

$$L = L_{DSC} + L_{CE} \tag{2}$$

**Strategies for using partially labeled and unlabeled data.** To obtain complete organ annotations that meet the training requirements, we use pseudo labels generated by the FLARE22 winning algorithm [8]. Specifically, for each training example of partially labeled data we replaced the missing organ labels in the ground truth with the organ labels provided by the pseudo-labels. For unlabeled data, we directly used the provided pseudo-labels.

### 2.3   Training strategy

The overall training strategy of our proposed method is as follows:
1. Train the organ segmentation model on all data obtained by strategy mentioned above.
2. Collect data containing tumor label and crop them to minimal box containing organ as training data to train the tumor segmentation model.
3. Generate tumor pseudo-label on unlabelled data using tumor segmentation model.
4. Combine data with ground truth label and data with pseudo label to train final tumor segmentation model.

### 2.4   Anatomical prior Post-processing

**Aorta-based cropping.** Due to the inclusion of non-abdominal organs such as the lungs and pelvis in a significant portion of the data, false segmentation of these organs, for example mistaking the bladder for the liver or stomach, can occur easily. To leverage anatomical prior knowledge and minimize false segmentation, we employed a cropping approach by defining the upper boundary

as the highest position of the aorta and setting the lower boundary as 20 layers below its lowest position. This strategy allows us to focus on abdominal organ segmentation while reducing errors.

**Tumor connectivity analysis.** In reality, tumors are typically connected to their corresponding organs rather than existing independently outside them. Although free tumor components may appear in segmentation results due to undersegmentation at junctions, we implemented an additional step to identify and remove disconnected tumor components from their respective organs. By doing so, we effectively mitigate false segmentation issues associated with free tumor structures.

### 2.5   Acceleration for inference

**Interpolating functions.** In the process of nnU-Net inference, the process of downsampling-inference-upsampling of the results is required, and we find that this part consumes a lot of time. Therefore, we replace the interpolation method of nnU-Net with the pytorch based interpolation function, which adopts the area mode for downsampling and the trilinear mode for upsampling.

**GPU acceleration.** We find that using the GPU during interpolation can greatly accelerate the computation, but due to the large size of most CT scans, this would take up a lot of GPU resources, making it impossible to run on more devices. Therefore, we adopted the cropping probability map-interpolation-merge process, which can accelerate the calculation while running in a smaller GPU occupancy.

## 3   Experiments

### 3.1   Dataset and evaluation measures

The FLARE 2023 challenge is an extension of the FLARE 2021-2022 [15][16], aiming to aim to promote the development of foundation models in abdominal disease analysis. The segmentation targets cover 13 organs and various abdominal lesions. The training dataset is curated from more than 30 medical centers under the license permission, including TCIA [3], LiTS [1], MSD [20], KiTS [6,7], autoPET [5,4], TotalSegmentator [21], and AbdomenCT-1K [18]. The training set includes 4000 abdomen CT scans where 2200 CT scans with partial labels and 1800 CT scans without labels. The validation and testing sets include 100 and 400 CT scans, respectively, which cover various abdominal cancer types, such as liver cancer, kidney cancer, pancreas cancer, colon cancer, gastric cancer, and so on. The organ annotation process used ITK-SNAP [22], nnU-Net [10], and MedSAM [14].

The evaluation metrics encompass two accuracy measures—Dice Similarity Coefficient (DSC) and Normalized Surface Dice (NSD)—alongside two efficiency

measures—running time and area under the GPU memory-time curve. These metrics collectively contribute to the ranking computation. Furthermore, the running time and GPU memory consumption are considered within tolerances of 15 seconds and 4 GB, respectively.

### 3.2    Implementation details

**Environment settings** The development environments and requirements are presented in Table 1.

**Table 1.** Development environments and requirements.

| | |
|---|---|
| System | Ubuntu 20.04.1 |
| CPU | Intel(R) Xeon(R) Silver 4216 CPU @ 2.10GHz |
| RAM | 16×4GB; 2.67MT/s |
| GPU (number and type) | Four Nvidia GeForce RTX 3090 24GB |
| CUDA version | 11.7 |
| Programming language | Python 3.9.17 |
| Deep learning framework | torch 2.0.1, torchvision 0.15.2 |
| Specific dependencies | nnU-Net 2.1.1 |
| Code | https://github.com/w58777/FLARE23 |

**Training protocols** The training protocols for the organ segmentation network and the tumor segmentation network are listed in Tables 2 and 3, respectively. During training, we used additive luminance transformation, gamma transformation, rotation, scale transformation, and elastic deformation for data augmentation.

## 4    Results and discussion

### 4.1    Quantitative results on validation set

The overall quantitative results are shown in Table 4. We performed ablation experiments on tumor segmentation to validate the effect of unlabeled data. Table 5 shows the results with or without the use of unlabeled data. It can be noticed that semi-supervised model outperforms fully supervised model using only labeled data. This is due to the fact that semi-supervised methods utilize unlabeled data which greatly enhance the generalization of model. This also confirms the data-driven of deep learning.

**Table 2.** Training protocols for organ segmentation.

| | |
|---|---|
| Network initialization | "He" normal initialization |
| Batch size | 2 |
| Patch size | 96×160×160 |
| Total epochs | 1000 |
| Optimizer | SGD with nesterov momentum ($\mu = 0.99$) |
| Initial learning rate (lr) | 0.01 |
| Lr decay schedule | Poly learning rate policy:$(1 - epoch/1000)^{0.9}$ |
| Training time | 26 hours |
| Loss function | Dice loss and cross entropy loss |
| Number of model parameters | 36.99M |
| Number of flops | 248G |
| $CO_2$eq | 7.8 Kg |

**Table 3.** Training protocols for tumor segmentation.

| | |
|---|---|
| Network initialization | "He" normal initialization |
| Batch size | 4 |
| Patch size | 56×112×176 |
| Total epochs | 1000 |
| Optimizer | SGD with nesterov momentum ($\mu = 0.99$) |
| Initial learning rate (lr) | 0.01 |
| Lr decay schedule | Poly learning rate policy:$(1 - epoch/1000)^{0.9}$ |
| Training time | 22.8hours |
| Number of model parameters | 48.88M |
| Number of flops | 291G |
| $CO_2$eq | 5.5 Kg |

**Table 4.** Quantitative evaluation results, the public validation denotes the performance on the 50 validation cases with ground truth, the online validation denotes the leaderboard results. All results are presented with the mean score and standard deviation of DSC and NSD.

| Target | Public Validation | | Online Validation | | Testing | |
|---|---|---|---|---|---|---|
| | DSC(%) | NSD(%) | DSC(%) | NSD(%) | DSC(%) | NSD (%) |
| Liver | 97.94 ± 0.48 | 98.88 ± 1.09 | 98.08 | 99.04 | | |
| Right Kidney | 96.33 ± 2.58 | 96.52 ± 4.15 | 94.26 | 95.47 | | |
| Spleen | 97.11 ± 2.63 | 98.26 ± 3.50 | 96.86 | 98.27 | | |
| Pancreas | 86.40 ± 5.23 | 96.56 ± 4.23 | 85.65 | 95.96 | | |
| Aorta | 96.43 ± 3.63 | 98.58 ± 3.44 | 97.36 | 99.26 | | |
| Inferior vena cava | 92.97 ± 4.62 | 94.00 ± 5.19 | 93.23 | 94.16 | | |
| Right adrenal gland | 84.68 ± 12.78 | 94.68 ± 13.78 | 85.84 | 95.81 | | |
| Left adrenal gland | 83.64 ± 5.76 | 95.45 ± 3.91 | 84.86 | 94.92 | | |
| Gallbladder | 86.07 ± 19.47 | 86.83 ± 20.62 | 86.55 | 86.97 | | |
| Esophagus | 81.68 ± 16.65 | 90.73 ± 16.99 | 83.87 | 93.26 | | |
| Stomach | 93.84 ± 4.09 | 97.01 ± 4.66 | 94.58 | 97.47 | | |
| Duodenum | 82.63 ± 7.72 | 94.53 ± 5.48 | 83.79 | 95.19 | | |
| Left kidney | 93.95 ± 11.06 | 94.21 ± 12.44 | 94.43 | 94.96 | | |
| Tumor | 42.87 ± 35.86 | 38.41 ± 32.76 | 41.89 | 36.14 | | |
| Average | 66.57 | 66.75 | 87.23 | 91.20 | | |

**Table 5.** Ablation experiments on tumor segmentation to validate the effect of unlabeled data.

| method | Organ DSC | Organ NSD | Tumor DSC | Tumor NSD |
|---|---|---|---|---|
| w/ unlabeled data | 90.28 | 95.10 | 42.87 | 38.41 |
| w/o unlabeled data | 89.44 | 94.97 | 41.89 | 34.76 |

**Table 6.** Quantitative evaluation of segmentation efficiency in terms of the running them and GPU memory consumption. Total GPU denotes the area under GPU Memory-Time curve.

| Case ID | Image Size | Running Time (s) | Max GPU (MB) | Total GPU (MB) |
|---|---|---|---|---|
| 0001 | (512, 512, 55) | 23.57 | 3192 | 18634 |
| 0051 | (512, 512, 100) | 21.74 | 4748 | 32704 |
| 0017 | (512, 512, 150) | 27.93 | 2298 | 38197 |
| 0019 | (512, 512, 215) | 27.32 | 2220 | 34973 |
| 0099 | (512, 512, 334) | 28.49 | 2278 | 38045 |
| 0063 | (512, 512, 448) | 35.54 | 2276 | 51162 |
| 0048 | (512, 512, 499) | 44.31 | 2248 | 58454 |
| 0029 | (512, 512, 554) | 47.68 | 2260 | 63090 |

## 4.2 Qualitative results on validation set

Examples of good segmentation and poor segmentation are given in Fig 2. The qualitative results show that our method performs well in segmenting organs such as liver, kidney, etc. Meanwhile, there are problems in recognizing and segmenting organs such as duodenum and adrenal gland. This may be due to the fact that large organs such as liver and kidney have more obvious boundaries in CT images, while some organs such as duodenum and adrenal gland are closely connected with other organs anatomically and have low contrast with their surroundings, making it difficult to separate them from the background and other organs. In addition, in tumor segmentation stage, our proposed algorithm can identify and segment liver tumors as well as kidney tumors, however, it performs poorly in segmenting giant tumors and boundary diffuse tumors.

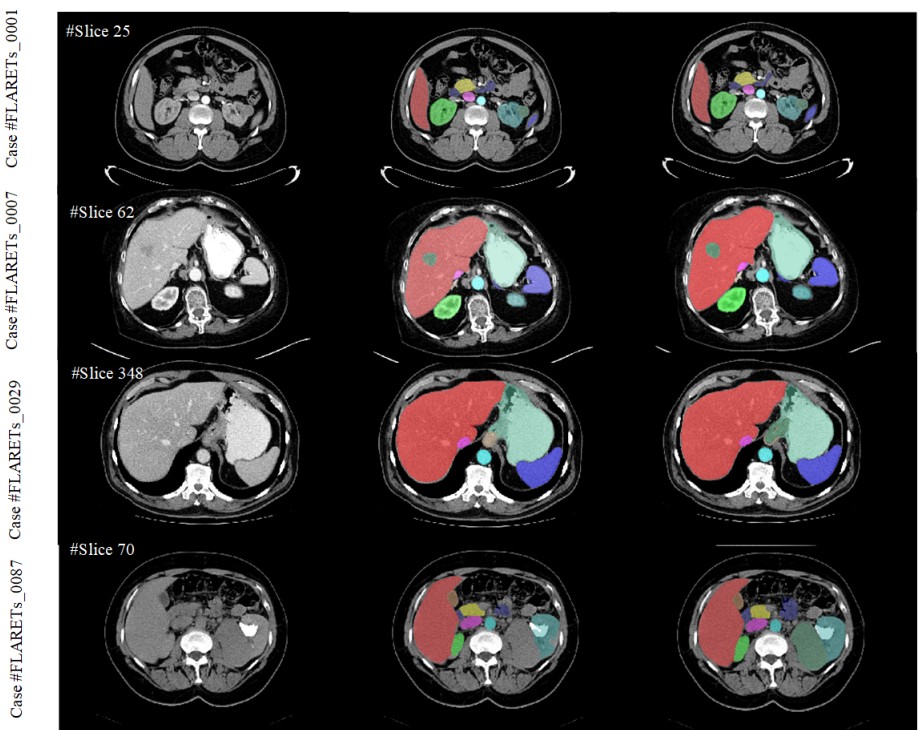

**Fig. 2.** Segmentation examples of good and poor cases. Our model performs well in segmenting most of the organs. At the same time, it has problems in segmenting small organs with low contrast and unusually large tumors.

### 4.3   Segmentation efficiency results on validation set

We ran our model on a docker with NVIDIA GeForce RTX 3090 (24G) and 28 GB RAM for inference on 100 validation cases. The average inference time per case is 23.77 s, the average maximum GPU memory used for inference is 2755.8 MB, and the average GPU- time AUC area under the curve is 29,484.24. Table 6 shows the inference efficiency parameters of our model on some examples.

### 4.4   Results on final testing set

This is a placeholder. We will send you the testing results during MICCAI (2023.10.8).

### 4.5   Limitation and future work

Qualitative and quantitative results show that our method performs well for most of organs segmentation, but for some small organs and tumors, our segmentation method is not robust enough. In addition, for cases with more CT slices, the abdominal region is difficult to extract and the segmentation efficiency is not satisfactory. Meanwhile, although the coarse-to-fine segmentation improves the accuracy of tumor segmentation, it also increases the inference time to some extent. Our future work will focus on the segmentation of small organs and tumors to develop more accurate segmentation algorithms for small targets.

## 5   Conclusion

In this study, we present a coarse-to-fine model for multi-organ and pan-tumor segmentation in abdominal CT. By using unlabeled data, our methods can improve segmentation performance. Our method also balances inference efficiency and segmentation accuracy to achieve accurate and fast multi-organ and pan-cancer segmentation. Quantitatively evaluated, our method achieves an average DSC of 90.28% on multi-organ and 42.87% on tumor, with an average process time of 23.77s per case in the validation dataset.

**Acknowledgements** The authors of this paper declare that the segmentation method they implemented for participation in the FLARE 2023 challenge has not used any pre-trained models nor additional datasets other than those provided by the organizers. The proposed solution is fully automatic without any manual intervention. We thank all the data owners for making the CT scans publicly available and CodaLab [19] for hosting the challenge platform.

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

**Table 7.** Checklist Table. Please fill out this checklist table in the answer column.

| Requirements | Answer |
| --- | --- |
| A meaningful title | Yes |
| The number of authors (≤6) | 3 |
| Author affiliations, Email, and ORCID | Yes |
| Corresponding author is marked | Yes |
| Validation scores are presented in the abstract | Yes |
| Introduction includes at least three parts: background, related work, and motivation | Yes |
| A pipeline/network figure is provided | Figure 3 |
| Pre-processing | Page 3 |
| Strategies to use the partial label | Page 4 |
| Strategies to use the unlabeled images. | Page 4 |
| Strategies to improve model inference | Page 5 |
| Post-processing | Page 4 |
| Dataset and evaluation metric section is presented | Page 5 |
| Environment setting table is provided | Table 1 |
| Training protocol table is provided | Table 2 and Table 3 |
| Ablation study | Page 8 |
| Efficiency evaluation results are provided | Table 6 |
| Visualized segmentation example is provided | Figure 2 |
| Limitation and future work are presented | Yes |
| Reference format is consistent. | Yes |