# OpenReview forum: "Semi-supervised Abdominal Multi-Organ and Tumors Segmentation by Cascaded nnUNet"
_MICCAI.org/2023/FLARE — Submitted to FLARE 2023_

### Official Review · Reviewer_iEJX · 2023-09-19
**Good job**

**Rating:** 8
**Confidence:** 4

**Review:**

Pros:

1）This article has high completeness
2)   Clear segmentation pipeline

Cons:
1) Inconsistent article layout，with indent in the Introduction but not in other parts
2) The Equation 1 maybe wrong

---

### Official Review · Reviewer_qhWa · 2023-09-19
**Semi-supervised Abdominal Multi-Organ and Tumors Segmentation by Cascaded nnUNet**

**Rating:** 8
**Confidence:** 4

**Review:**

Strengths:
- The proposed method achieves accurate and efficient segmentation of abdominal organs and tumors. The method achieves an average DSC of 90.28% on abdominal multi-organ segmentation and 42.87% on pan-tumor segmentation, with an average inference time of 23.77s per case on validation set.

Weaknesses:
- Change “nnUNet” to “nnU-Net” in title and Fig.1.
- The quantitative results of NSD and area under the GPU memory time curve are not presented in abstract.
- In Sec. 2.2, Paragraph 1, delete one “two” in “Our method composes of two two 3D nnU-Net”.
- The description of “organ minimum box” is not clear. Does it mean minimum box of one organ or minimum box of all segmented organs? Does the organ minimum box corresponds to one organ or abdominal region?
- In aorta-based cropping, how do you find the highest position and the lowest position of the aorta?
- How do you design the two experiments in ablation study on utilization of unlabeled data?
- Please check average results on public validation set in Table 4.
- Change “running them” to “running time” in annotation of Table 6.
- Please check grammar.

---

### Official Review · Reviewer_uZPb · 2023-09-19
**Semi-supervised Abdominal Multi-Organ and Tumors Segmentation by Cascaded nnUNet**

**Rating:** 7
**Confidence:** 4

**Review:**

Pros:
1. The article has a complete structure.
2. The method obtains an average DSC of 90.28% on abdominal multi-organ segmentation and 42.87% on pan-tumor segmentation, with an average inference time of 23.77s per case on the validation set.


Cons:
1. The average results on public validation set in Table 4 are somewhat strange.
 2. There are no quantitative results of NSD and area under the GPU memory time curve in the abstract.
3. The description of Aorta-based cropping is not clear.

---

### Official Review · Reviewer_bjYt · 2023-09-20
**Semi-supervised Abdominal Multi-Organ and Tumors Segmentation by Cascaded nnUNet**

**Rating:** 7
**Confidence:** 4

**Review:**

Pros:
 1. Well-structured articles
 2. The method proposes a cascaded nnU-Net approach and segments tumors of different organs separately to improve the accuracy of tumor segmentation. And the inference speed is improved by post-processing. Finally, the organ DSC 90.28% as well as the tumor DSC 42.87% were obtained. The inference time was 23.77s.

Cons：
 1. The title should be nnU-Net not nnUNet.
 2. 2.2 seems to have a typographical error, it should read two 3D nnU-Net.
 3. Did the experiment use labeled data, meaning data with tumor labels, or was all of the partially labeled data utilized in its entirety?
 4. The average results for the public validation set in Table4 appear to be wrong.
 5. The qualitative results visualization graph does not show ablation comparison experiments.

---

> ### Comment · Reviewer_bjYt · 2023-11-23
> **Second round of review**
>
> Doubts in the paper were not revised by the authors and the authors still need to revise the content to meet the publication requirements

---

### Official Review · Reviewer_pcsv · 2023-10-03
**Good writing, well structured with minor issues**

**Rating:** 7
**Confidence:** 5

**Review:**

A cascaded framework is proposed for abdominal organ and tumor segmentation in FLARE challenge 2023. Images with pseudo labels provided by the organizers were used for training an organ segmentation model, which is followed by a tumor segmentation model. The writing is overall smooth and clear.  Some issues can be addressed to improve the quality:
1)	The definition of z-score in equation 1 is not correct. it should be (x – u)/sigma.
2)	For the aorta-based cropping, how to get the aorta region for unlabeled images and testing images where the labels of aorta were not available?
3)	Page 5, ’20 layers’ should be ’20 slices’
4)	For the dataset, it would be better to provide more information about the image size and spatial resolution.
5)	For ablation study, the effectiveness of the cascaded framework is not shown. Is there any improvement from not using the cascaded framework?

---

### Official Review · Reviewer_MAzo · 2023-10-04
**A cascaded approach using cascaded nnU-Net**

**Rating:** 8
**Confidence:** 5

**Review:**

Pros:
1. Complete structure;

Cons:
1. Some important preprocessing paramters are not introduced, such as target spacing;
2. The coarse-to-fine strategy seems only used for tumor, but not for organs?

---

### Decision · Program_Chairs · 2023-10-24

**Decision:**

Reject

**Comment:**

The authors didn't make responses to the valuable review comments.